# Students' Preferences and Perceptions Regarding Online versus Offline Teaching and Learning Post-COVID-19 Lockdown

Fatima Riaz [1], Syed Esam Mahmood [1,*], Tahmina Begum [2], Mohammad Tauheed Ahmad [3], Ayoub A. Al-Shaikh [1], Ausaf Ahmad [4], Ayed A. Shati [5] and Mohammad Suhail Khan [6]

1   Department of Family and Community Medicine, College of Medicine, King Khalid University, Abha 62529, Saudi Arabia
2   Department of English, College of Science and Arts, King Khalid University Tanomah, Abha 62529, Saudi Arabia
3   Department of Medical Education, College of Medicine, King Khalid University, Abha 62529, Saudi Arabia
4   Department of Community Medicine, Integral Institute of Medical Science and Research, Integral University, Kursi Road, Lucknow 226026, India
5   Department of Child Health, College of Medicine, King Khalid University, Abha 62529, Saudi Arabia
6   Department of Public Health, College of Applied Medical Sciences, Khamis Mushait, King Khalid University, Abha 61412, Saudi Arabia
*   Correspondence: smahmood@kku.edu.sa; Tel.:+966-550484344

**Abstract:** The COVID-19 pandemic at its peak compelled students to stay home and adapt to the distance learning system. The world has gone through phases of fear and respite in the recent years. There have been a number of studies related to student learning via online teaching during the pandemic. Now, as the vaccination coverage picks up and the pandemic appears to have achieved a plateau, it is time to take a view of students' perceptions of online learning and its effectiveness in skill development during the COVID-19 pandemic. This study assesses the students' preferences and perceptions regarding offline and online learning post-COVID-19 lockdown and with the resumption of offline classes. A cross-sectional study was conducted at King Khalid University, Aseer region, from the period of 1 January 2022 to 30 January 2022. A convenience sampling technique was utilized to collect data from female students. Data analysis was conducted by using SPSS version 22.0. A total of 480 students participated in the study, their mean age was $19.79 \pm 1.48$. More than half of the students (64%) still fear getting COVID-19 as they continue with the in-person or offline classes despite having completed their doses of vaccination. Almost half of the students have difficulty in waking up after the recommencement of offline classes. The majority (77%) felt tired after starting offline classes, and 63% felt unhappy after starting offline classes again. The majority of students believe that, with offline classes, they have difficulty in time management and concentration. The majority of students believed that, with online classes, they are more comfortable in gaining knowledge and learning, more alert, more satisfied, and gain higher scores in exams. The majority of students preferred the online mode of learning, with about 72% of students wishing to continue online learning in the future. This research underlines the influence of fear of getting COVID-19 after the commencement of face-to-face learning amongst the students. Students were more inclined to continue with online classes because of fear of getting COVID-19, despite having full doses of the vaccination in Saudi Arabia. There is a need for a better understanding of students' motivations and coping mechanisms during the pandemic.

**Keywords:** COVID-19; online; offline; learning

## 1. Introduction

Since the emergence of the COVID-19 pandemic, the whole world has been transformed, bringing tremendous changes in our lives [1]. In December 2019, an outbreak of pneumonia-like illness originated in the city of Wuhan in the Peoples Republic of China,

genomic analysis confirmed that the illness was related to severe acute respiratory syndrome coronavirus 2 (SARS-CoV2). It not only causes pneumonia but also a continuum of respiratory tract symptoms such as difficulty in breathing, fever, and flu-like symptoms [2]. Because of the rapid spread of this virus and its illness leading to millions of deaths all over the world, the World Health Organization declared on 12 March 2020 the COVID-19 pandemic. To prevent mortalities, lockdown has been imposed in many countries to facilitate strict social distancing so that further public health catastrophe can be minimized [3].

The COVID-19 pandemic has imposed a vast disruption in every aspect of our lives, such as daily living, physical health, mental health, office work, industries, agriculture, politics, health sector, tourism, and even education. It has had a devastating effect on human life not only in terms of mortality and morbidity but also in terms of the psychosocial and socioeconomic impact. It has imposed disruption to the larger canvas of the education system which has affected almost 1.6 billion students globally. Closure of educational institutions due to lockdown and maintaining social distancing affected almost 94% of the students all over the world. Social distancing and restrictive movement policies have significantly disturbed traditional educational practices all over the world. These restrictions have been associated with unprecedented challenges, not only for students, but also for teachers due to the abrupt transition to online learning. It was very challenging to adopt another learning system that could replace face to the face learning system [4]. It was believed that distant teaching and learning experience was of less quality than the quality of teaching in the traditional classrooms; many past studies showed the negative attitude of students during video conferencing and distance learning, but the COVID-19 pandemic changed all concepts and attitudes upside down [5].

In the recent pandemic, there was no choice to avoid online learning systems so they became the alternative system of education all over the world. Face-to-face learning activities are mandated not to be carried out, following standard operating procedures (SOPs) [6].

Therefore, in the education sector, the online mode of learning is adopted widely within a very short period when it was not practiced before in many subjects at all. This system has been adopted to prevent losing academic year accomplishments and learning goals because, with a face-to-face learning, there was a great chance of further spreading the COVID-19 infection. For some faculties of studies, the online system was new, and conducting online teaching and learning to acquire the best outcome was a great challenge for all [7].

Different studies indicated that both students and faculty were not well prepared or equipped for such a quick transition. About 62% of students were underprepared to deal with the online learning system. However, the system has to be implemented to prevent the further spread of the pandemic [8].

All that agony is over now and the lockdown of institutions due to the COVID-19 pandemic has been lifted, but the period which we all suffered still imposes fear of getting COVID-19 even after full doses of vaccination [9].

Schools and other academic institutes are reopened at present, after the lifting of lockdown, presenting the challenges of regaining face-to-face learning goals and implementing the system. The pandemic forced students to stay home and adapt to the distance learning system. However, students' motivation for online learning and its effectiveness in skill development during the COVID-19 pandemic has not been widely studied. Studying the perceptions and preferences of the students regarding virtual learning as compared to face-to-face learning can help develop a curricular approach which integrates the best elements of offline as well as online learning, in service to better learning outcomes. The findings and implications of this study, which can be explained by different educational and public health theories, e.g., self-efficacy theory, self-determination theory, social constructivism, theory of planned behavior, etc. These may be helpful in devising the future strategy for curriculum delivery with the optimal integration of elements of virtual learning and face-to-face learning into the curricular scheme, in service to the best academic outcomes.

This study was undertaken to assess the students' preferences and perceptions regarding the post-COVID-19-lockdown resumption of face-to-face classes.

## 2. Methodology

### 2.1. Study Area and Population

A cross-sectional study was conducted at King Khalid University, Aseer region from the period of 1 January 2022 to 30 January 2022. A convenient sampling technique was utilized to collect data from female students. A total of 500 forms were distributed among students. A total of 480 students returned the completed forms.

We hypothesized that students continue to prefer the online learning post-COVID-19 lockdown resumption of face-to-face classes. The research question was: What are the students' preferences and perceptions regarding online versus offline learning after the resumption of face-to-face classes post-COVID-19-lockdown?

### 2.2. Inclusion and Exclusion Criteria

All students from science and arts departments who have completed vaccination against COVID-19 and who were willing to participate were included in the study. Those who were not vaccinated for COVID-19 or were not willing to participate were excluded from the study. The students' confidentiality and anonymity were completely maintained.

### 2.3. Ethical Approval, Questionnaire and Data Collection

This study was approved by the Research Ethical Committee of the College of Medicine, King Khalid University. A 15-item self-constructed questionnaire was utilized for data collection. It consisted of closed ended questions. The questionnaire was developed by the researchers based on the students' feedbacks regarding the online and offline teaching and learning and further reviewed by experts for validity and applicability.

Here we are concerned with the mode of learning and teaching by which we deliver lectures; for example, the traditional learning, i.e., face-to-face, method (offline), online learning, and the blended method, which means both offline and online mode of learning and teaching [10]. Online learning is described as access to learning experiences via the use of some technology [11]. It has been identified as a version of distance learning which improves access to educational opportunities for both nontraditional and disenfranchised learners [12,13].

Before and after lockdown, the (offline) traditional mode of learning and teaching was implemented in the addressed institution. During lockdown, there was a sudden shift from traditional learning to online learning.

The information included the students' age and subjective fear of COVID-19 after starting offline classes. The impact of offline classes on the students' happiness and tiredness were measured only by the subjective feeling of students, ascertained via questions related to their perception regarding the method of learning and future choices of learning methods. A pilot study was conducted which consisted of 30 questions. This initial data was analyzed but not included in the results. The questionnaire was then modified and validated for data collection. Data was collected by the principal investigator herself after taking informed verbal consent and by ensuring students that their identities were kept confidential.

### 2.4. Data Entry and Analysis

Data were collected and entered by the first author herself. Analysis was conducted using SPSS version 22. For continuous variables, mean and standard deviation were used. For the relationship between fear of getting COVID-19 after the commencement of offline classes and the learning methods and their impact, and the relationship between perceptions and fear of getting COVID-19 and resuming offline classes among students after lockdown, a chi-square test was applied. A $p$-value of $<0.05$ was considered statistically significant.

### 3. Results

A total of 480 college students participated in the study; their mean age in years was 19.79± 1.48.

Figure 1 illustrates that almost 2/3rd of the students (64%) still fear getting COVID-19 after starting offline classes despite having completed vaccine doses.

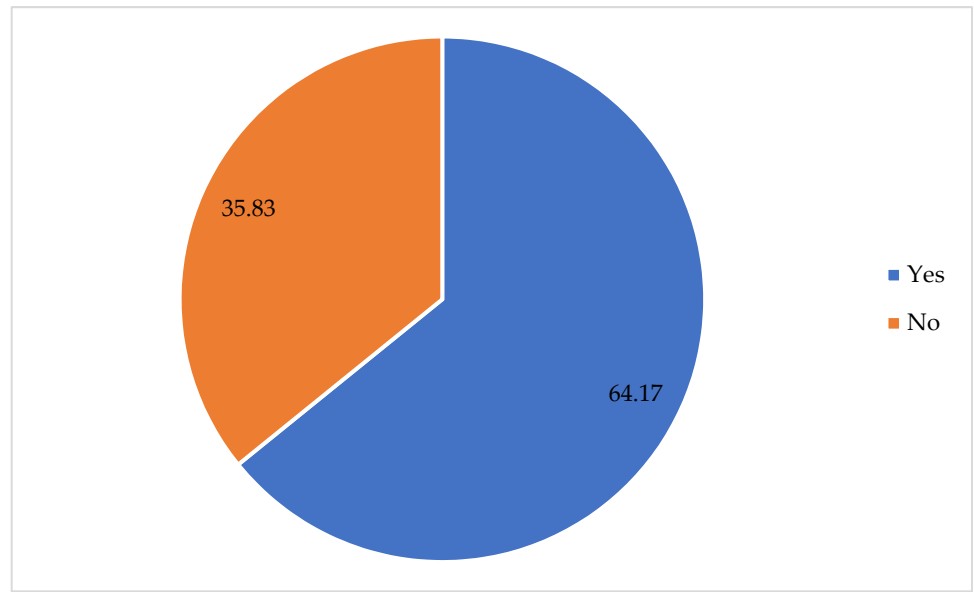

**Figure 1.** Fear of getting COVID-19 after starting offline classes.

Figure 2 shows the impact of starting offline classes among students after the COVID-19 lockdown. Almost half of the students (50.83%) reported having difficulty in waking up after the recommencement of offline classes. The vast majority of students (77%) reported feeling tired after starting offline classes, and the majority of students (63%) were unhappy after starting offline classes again.

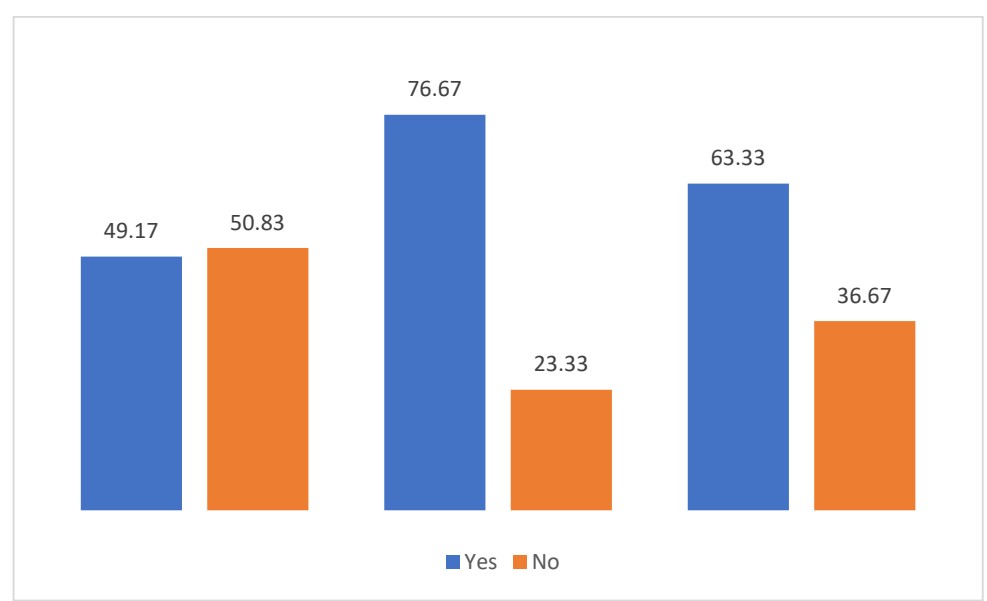

**Figure 2.** Impact of resuming back offline classes on students after COVID-19 lockdown.

Table 1 shows the students' perceptions regarding different teaching and learning methods. About half of the students (52.5%) believe that, with offline classes, they have

difficulty in time management, and 57.5% of students have difficulty in concentration in offline classes. About 70.8% of students believe that, with online learning, they were more comfortable with for acquiring knowledge. More than half of the students (59.2%) believe that online learning method keeps them more alert and the vast majority (83.3%) of students are more satisfied with online learning and think online methods help them in gaining good scores in exams. Around 85.5% of students prefer online learning and 71.7% want to pursue online methods in future studies. Around 45.8% of students faced difficulties in getting drivers to reach to the college and more than half (56.7%) of the students believe that offline learning method keeps them more active.

**Table 1.** Students' perceptions regarding teaching/learning methods and their impact (*n* = 480).

| Variables | Frequency | Percentage |
|---|---|---|
| **Learning method has difficulty in time management** | | |
| Online | 228 | 47.5 |
| Offline | 252 | 52.5 |
| **Learning method has difficulty in concentration** | | |
| Online | 204 | 42.5 |
| Offline | 276 | 57.5 |
| **Learning method makes you more comfortable for gaining knowledge** | | |
| Online | 340 | 70.8 |
| Offline | 140 | 29.2 |
| **Learning method keeps you more alert** | | |
| Online | 284 | 59.2 |
| Offline | 196 | 40.8 |
| **Learning method satisfies you more** | | |
| Online | 400 | 83.3 |
| Offline | 80 | 16.7 |
| **Learning method you like most** | | |
| Online | 412 | 85.8 |
| Offline | 68 | 14.2 |
| **The method helps you obtain high scores in exams** | | |
| Online | 400 | 83.3 |
| Offline | 80 | 16.7 |
| **The method helps you in learning more** | | |
| Online | 328 | 68.3 |
| Offline | 152 | 31.7 |
| **Learning method you want to continue in future** | | |
| Online | 344 | 71.7 |
| Offline | 136 | 28.3 |
| **Difficulty in getting a driver to reach to the college** | | |
| Yes | 220 | 45.8 |
| No | 260 | 54.2 |
| **Learning method keeps you more physically active** | | |
| Online | 208 | 43.3 |
| Offline | 272 | 56.7 |

Figure 3 illustrates that the students' preferences regarding online modes of learning compared to offline, with about 72% of students wishing to continue online learning in the future. About 83.30% of students were more satisfied with online modes of learning and believe that online the method helps them in getting good scores in exams. Around 85.80% of students like the online method most; whereas only 14.20% of students like the offline method most.

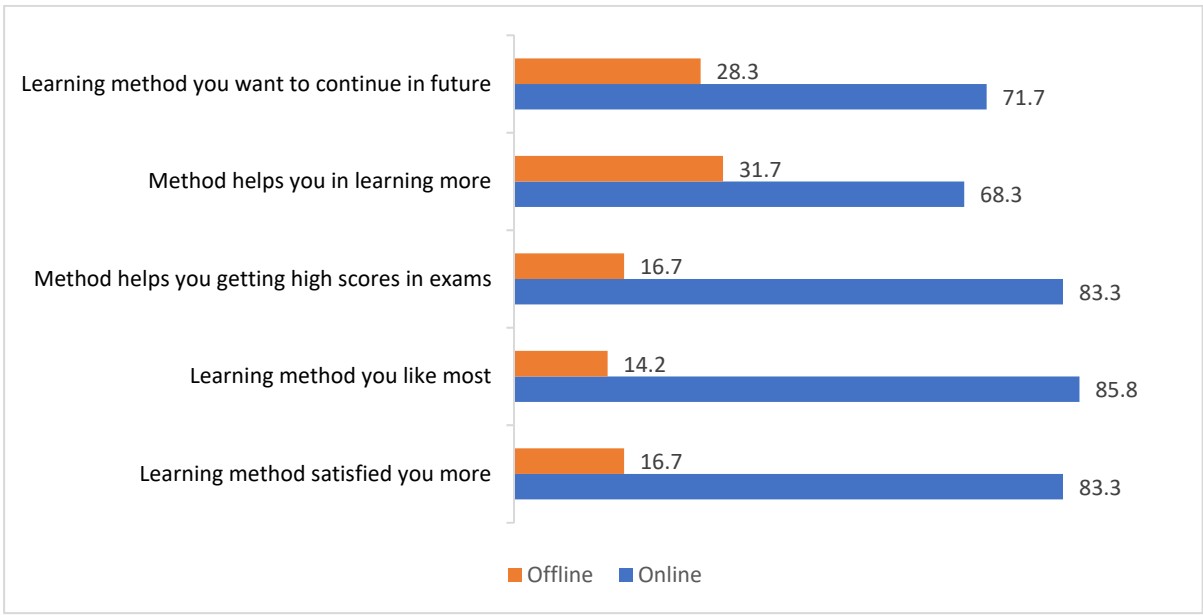

**Figure 3.** Perceptions regarding teaching/learning methods.

Table 2 demonstrates the influence of fear of getting COVID-19 on the impact of resuming offline classes among students after lockdown. Among those students who fear getting COVID-19, *p*-values were significant for difficulty in waking up after starting online classes, feeling tired after starting offline classes, and feeling unhappy after starting offline classes.

**Table 2.** Relationship between perceptions and fear of getting COVID-19 among students on resumption of offline classes post lockdown.

| Characteristics | Fear of Getting COVID-19 after Starting Offline Classes | | Chi-Square | *p*-Value | OR | 95%CI | |
|---|---|---|---|---|---|---|---|
| | Yes | No | | | | Lower | Upper |
| **Difficulty in waking up after starting offline classes** | | | | | | | |
| Yes | 176(74.6) | 60(25.4) | 21.87 | **0.000 *** | 2.49 | 1.69 | 3.66 |
| No | 132(54.1) | 112(45.9) | | | | | |
| **Feel tired after starting offline classes** | | | | | | | |
| Yes | 272(73.9) | 96(26.1) | 65.15 | **0.000 *** | 5.98 | 3.78 | 9.47 |
| No | 36(32.1) | 76(67.9) | | | | | |
| **Feeling unhappy after starting offline classes** | | | | | | | |
| Yes | 228(75) | 76(25) | 42.31 | **0.001 *** | 3.60 | 2.43 | 5.34 |
| No | 80(45.5) | 96(54.5) | | | | | |

*p*-value of < 0.05 statistically significant *.

Table 3 shows the influence of fear of getting COVID-19 on learning method choices among students. *p*-values were also significant for those who believe they have difficulty in concentrating with offline classes, who are comfortable in gaining knowledge with online method, who are more satisfied with the online method, who believe online learning keeps them more alert, who prefer the online method to the offline method, who believe that the online method helps them to learn more, who wanted to pursue the online method of learning in the future, and who have difficulty in getting drivers to reach to the college. *p*-values were not significant for those who believe they have difficulty in time management with online learning, who believe the online method helps them in getting high scores in exams, and who believe offline physical activity keep them active.

**Table 3.** Relationship between perceptions and fear of getting COVID-19 on learning preferences among students (*n* = 480).

| Perceptions | Fear of Getting COVID-19 after Starting Offline Classes | | Chi-Square | *p*-Value | OR | 95%CI | |
|---|---|---|---|---|---|---|---|
| | **Yes** | **No** | | | | **Lower** | **Upper** |
| **Learning method has difficulty in time management** | | | | | | | |
| Online | 152(66.7) | 76(33.3) | 1.18 | 0.277 | 1.23 | 0.85 | 1.79 |
| Offline | 156(61.9) | 96(38.1) | | | | | |
| **Learning method has difficulty in concentration** | | | | | | | |
| Online | 104(51) | 100(49) | 26.82 | **0.000 *** | 0.37 | 0.25 | 0.54 |
| Offline | 204(73.9) | 72(26.1) | | | | | |
| **Learning method makes you more comfortable in gaining knowledge** | | | | | | | |
| Online | 228(67.1) | 112(32.9) | 4.241 | **0.039 *** | 1.53 | 1.02 | 2.29 |
| Offline | 80(57.1) | 60(42.9) | | | | | |
| **Learning method keeps you more alert** | | | | | | | |
| Online | 196(69) | 88(31) | 7.108 | **0.008 *** | 1.67 | 1.14 | 2.44 |
| Offline | 112(57.1) | 84(42.9) | | | | | |
| **Learning method Satisfied you more** | | | | | | | |
| Online | 272(68) | 128(32) | 8.394 | **0.004 *** | 2.60 | 1.59 | 4.23 |
| Offline | 36(45) | 44(55) | | | | | |
| **Learning method you like most** | | | | | | | |
| Online | 292(70.9) | 120(29.1) | 56.89 | **0.000 *** | 7.91 | 4.34 | 14.30 |
| Offline | 16(23.5) | 52(76.5) | | | | | |
| **The method helps you obtain high scores in exams** | | | | | | | |
| Online | 260(65) | 140(35) | 0.725 | 0.395 | 1.24 | 0.76 | 2.03 |
| Offline | 48(60) | 32(40) | | | | | |
| **The method helps you in learning more** | | | | | | | |
| Online | 232(70.7) | 96(29.3) | 19.41 | **0.000 *** | 2.42 | 1.62 | 3.59 |
| Offline | 76(50) | 76(50) | | | | | |

**Table 3.** *Cont.*

| Perceptions | Fear of Getting COVID-19 after Starting Offline Classes | | Chi-Square | *p*-Value | OR | 95%CI | |
|---|---|---|---|---|---|---|---|
| | **Yes** | **No** | | | | **Lower** | **Upper** |
| **Learning method you want to pursue in future** | | | | | | | |
| Online | 240(69.8) | 104(30.2) | 16.56 | **0.000 *** | 2.31 | 1.54 | 3.47 |
| Offline | 68(50) | 68(50) | | | | | |
| **Difficulty in getting drivers to reach to college** | | | | | | | |
| Yes | 156 (70.9) | 64(29.1) | 8.03 | **0.000 *** | 1.73 | 1.18 | 2.53 |
| No | 152(58.5) | 108(41.5) | | | | | |
| **Learning method keeps you more physically active** | | | | | | | |
| Online | 124(59.6) | 84(40.4) | 3.30 | 0.068 | 0.706 | 0.48 | 1.03 |
| Offline | 184(67.6) | 88(32.4) | | | | | |

*p*-value of <0.05 statistically significant *.

## 4. Discussion

The role of online teaching and learning cannot be underestimated; in fact, it has gained a lot of importance in this era, especially after the COVID-19 lockdown. The COVID-19 pandemic drastically caused all educational institutions all over the globe to close and shift their traditional mode of learning towards online methods which gave rise to multiple challenges for both students and teachers. Students faced some problems which influenced their behaviors towards online learning, for multiple reasons [14]. This is not only true of students; institutions also faced challenges in effectively engaging students' concentration over virtual learning platforms [15].

A rapid shift to online learning was mandated in Saudi Arabia to ensure social distancing and mitigate the spread of the COVID-19 pandemic. This mode of learning was carried further in view of the lingering COVID-19 situation which had caused extensive morbidity and mortality worldwide. With the invention of vaccination and strict measures regarding SOPs, control of coronavirus spread was achieved. Therefore, many countries lifted strict lockdown gradually and life gradually returned to normalcy. The mode of teaching and learning also shifted back to the traditional in-person mode.

In our study, we studied the impact of shifting back from online classes to face-to-face classes on students. This study finds that students in offline mode have less concentration comparing to face-to-face mode. This can be attributed to their fear of getting infected in classrooms. However, in-classroom presentation should help students concentrate more than online learning where students are at home with many distractions such as other family members, kids, etc. A majority of students (64%) are still afraid of getting COVID-19 despite having full doses of vaccination because of exposure due to face-to-face classes. One reason could be doubts regarding the effectiveness of the COVID-19 vaccine itself which was hurriedly launched and administered in urgency to the whole world. Some sections in the scientific community also have cast doubts on the efficacy and safety of the vaccine because of the rapid development and hurried approval of the vaccine [16]. In a study conducted amongst Malaysian students during the first-wave lockdowns, fear of contracting COVID-19 infection has been found as a major mental health issue apart from anxiety and depression [17]. A study conducted at Prince Sultan medical city at Riyadh also showed that almost 33% of their students fear getting COVID-19 while being at college and 57% of students think that they will bring COVID-19 into their home from college [18]. Fear of getting COVID-19 infection as a reason for continuing with the online classes even after the restrictions were lifted can be understood from the fact that wearing masks was still

compulsory in closed spaces as a precautionary measure. Distancing was still considered a desirable practice and hence the students might have been feeling more inclined to avoid any exposure. This can be explained by the theory of planned behavior, in which the intention to act in a particular way is affected by attitudes towards the behavior and its considered usefulness [19]. Moreover, the self-efficacy around the successful navigation of teaching/learning over a significant period post-implementation of the restriction could be another factor behind considering the exposure required to attend face-to-face classes as unnecessary [20].

Our study showed that almost half of the students have difficulty in waking up after commencement of offline classes again, 77% were feeling tired after starting offline classes, and 63% of students were unhappy after starting offline classes again. The reason might be that, in the absence of face-to-face classes, their activity level had decreased as well as their physical fitness and that is why they feel tired and unhappy after starting offline classes. Online classes during the COVID-19 pandemic have been found to be more time-efficient by students in a survey conducted in Saudi Arabia. A shift to offline classes might have confronted the student with initial challenges of time-management [21]. Another reason could be difficulty in breathing because of the mask, especially as this college is situated at a high altitude with lower levels of oxygen. In a survey of students before and after the first wave lockdown in UK, it was found that more students were experiencing sleep difficulties as well as decrease in concentration after the advent of the COVID-19 pandemic [22]. Another aspect of this issue is that students may find it difficult to manage their schedule in the event of a switch from offline to online and vice-versa. Hence, development of time-management skills has been identified as vital for any blended learning exercise [23].

Our study showed that more than half of our students also believe that, with offline classes, they feel more active than online classes because for attending lectures they have to get up and go to college and within the college they have to move from one place to another place by walking and that is why they feel more active during face-to-face classes. In other studies, it has been reported that COVID-19 institutional closures had led to an increased rate of obesity among children. During online classes during lockdown of institutions due to COVID-19 pandemic, students used to take lectures online from home so they have to sit at one place for hours which makes them inactive and as a result they gain weight. [23,24]. Schools and other educational institutes provide opportunities for physical activities and multiple programs facilitate and promote physical activity through physical education and sports [25–27]. Our findings are supported by a US study where, as a result of online classes, students had reduced levels of physical activities or even no physical activity which might have made them lethargic and unable to manage waking up for classes [28]. Nearly half (45.8%) of the students in our study have difficulty in getting drivers to reach the college. The lockdown in response to the COVID-19 pandemic resulted in remarkable job losses all over the globe [29]. All kind of employees related to every sort of profession kept losing their jobs. Transportation-related workers were more likely to be unemployed because of the lockdown of institutions due to COVID-19 pandemic and drivers who were working in customer-oriented sectors lost their jobs in even greater numbers [30]. The taxi journeys remained below half the pre-pandemic level in many countries [31]. This suggests that people who were dependent on drivers and taxis for their movement must be suffering because of lack of facilities. In Saudi Arabia, although females have started driving, the majority of females are still dependent on drivers; that is why female students have difficulty in getting drivers to reach to the college [32].

A majority of the students in our study were found unhappy after starting offline classes. This might be in continuation of the lockdown-related depression, loneliness, sadness, and anxiety which has been documented by different researchers [33]. The rate of psychological disorders increased substantially, and mental health was also affected badly during the lockdown of institutions due to COVID-19 pandemic. That could be a reason for lack of excitement after the reopening of the college. In our study, almost 70% of students liked online teaching and learning; these findings were consistent with another

study demonstrating that 75% of Australian students preferred recorded zoom lectures and more than half of Australian students did not want face-to-face interaction during their learning [34]. In another study conducted in Italy, students were also satisfied with distant leaning [35]. Another study conducted in Nigeria reflected that 47.7% of the students were satisfied with online teaching [36]. Almost 70% of our students think that online learning method helps them more towards good learning, similar findings were found in another study conducted in Lebanon [37].

Our study showed that the majority of students were comfortable, satisfied, and happy with online learning. This is similar to the findings of another survey conducted in Saudi Arabia where the students preferred online teaching/learning as it improved time-efficiency [21]. Apart from the ease offered by online learning, students also feel that they had acquired relatively better levels of expertise by using different online systems such as BlackBoard$^{TM}$, WebBoard$^{TM}$, or other online learning tools which they were not using during face-to-face learning. Students who have acquired skills to cope with these learning techniques would be more likely to see these systems as a way to expand their learning and the online system would facilitate higher self-efficacy levels than that of the offline system for them [38]. Students have more positive expectations from online learning systems outcomes, and consequently feel that they had acquired a relatively greater level of expertise and competency level with online learning. These beliefs make them inclined towards online learning [39,40]. Students at the university level in Saudi Arabia are generally familiar with operating the online learning gadgets; hence, they might feel comfortable navigating the learning management system and attending their sessions. Students of the present era are also called digital natives; they are already more advanced in utilizing social media, and all sorts of digital media more efficiently than the previous generations which is also one of the major factors for the preference of the online mode of learning [40,41]. This preference for online learning in this situation can be explained by a number of theories of learning and behavior. Successful experience with the online learning system which was imposed by the COVID-19 restrictions might have an effect of increased confidence pertaining to the use of online learning as explained by Bandura's self-efficacy theory [42]. Online learning does not require the need for restricted attendance in college and the need to prepare for that attendance which could be daunting for many. The exposure to online learning might have given the learners an enhanced sense of autonomy, which is an important aspect of motivation to learn as explained by self-determination theory [43].

Self-directed learning by which the learner takes control of the techniques and approach to learning is one of the important principles of Andragogy, or adult learning theory. In the online learning format, the learner could feel a greater sense of control about how they learn and manage their time and resources as per their requirements [44].

Advances in technology, including those pertaining to the internet, have favored the delivery of online education systems across the world during this pandemic and online learning grew at an incredible rate. Online education offers a more convenient and flexible way of learning than that of traditional face-to-face classes so students can balance their other activities, work, and family [45].

The COVID-19 pandemic has increased the importance of online teaching and learning in both developed countries and developing countries [46–49]. Although one study conducted in Pakistan showed that about 77% of medical students do not want to continue online learning in the future; in our study, almost 70% of our students want to pursue online learning in the future as well [50]. A study conducted in Poland among students showed 35.5% willingness regarding continuation of distant learning [51]. Online learning was experimented with before COVID-19 in different institutions and different subjects with variable degrees of acceptability from students and teachers [52,53].

Due to the ongoing COVID-19 pandemic, our students have developed a fear towards the unhealthy effects of offline teaching—they now feel tired and unhappy with face-to-face

classes—these findings are consistent with the existing literature on COVID-19 associated with students' well-being [54]. It has created a new era of research as well [17,55].

Overall, the fear of getting COVID-19 infection after the initiation of face-to-face classes influences students towards the continuation of online classes in the future. In some studies, students have reported showing better cognitive and motivational outcomes with the online mode of learning compared with face-to-face learning [56,57]. They were displaying a positive attitude towards the online system's adoption in future learning; however, before implementation, further research will be needed which should not be influenced by any fear. On the contrary, the blended learning method in which there are laboratory activities and interaction between classmates and teachers has been recommended by recent studies [58–60]. Moreover, a serious limitation of online learning is a deficiency of social context and diminished interaction between learner–learner and teacher–learner. While designing the online components, the problem-solving orientation to learning should be emphasized, as adult learners learn better when the learning is contextualized. This also relates to the constructivism theory which emphasizes the development of the capacity to solve real life problems based on one's existing understanding of the world and related concepts. The role of teachers is seen more efficiently as that of a guide rather than that of a provider of facts and concepts to be consigned to memory. Moreover, it is also important to remember that social–cultural context plays an important role in learning as postulated by Vygotsky. Learner–learner as well as teacher–learner interactions play an important role in the learning process; hence, ways of collaborative problem solving must be integrated into any learning modality and especially so while designing online components [61]. Judicious use of instructional design principles can help overcome this problem to a great extent. Moreover, it may be re-emphasized that further epidemiological studies are required to assess the perceived differences between online learning only and a blended learning approach.

## 5. Strengths and Limitations

The study findings are not intended to provide a generalization about remote learning in Saudi Arabia and point out the need for a deeper understanding of the state of student motivation and coping mechanisms during a pandemic. A small number of students were selected from two colleges which cannot represent the whole population. However, this study can serve as a baseline of students' views regarding learning methods and their impact after lifting lockdown and resuming previous learning methods. As increasing number of universities are taking initiatives to streamline the traditional educational system, incorporating elements of synchronous as well as asynchronous virtual learning into the curriculum. This study, which provides a window of insight into the preferences and perceptions of students on this issue, can help the academicians and administrators to develop optimum systems of curriculum delivery. This study may also help universities develop guidelines by which to effectively engage and ensure students' social presence in an online learning environment when facing an uncertain situation such as a pandemic. The study's results emphasize that students' online learning success and active learning are highly dependent on motivation and cognitive skills.

## 6. Conclusions

In this study, it is found that a significant majority of the students fear getting COVID-19 after the resumption of offline classes despite having a vaccination against COVID-19. They have difficulty in time management and have a feeling of tiredness when they attend the college in the face-to-face mode. The majority of them were not happy with the resumption of in-person classes and they wish to pursue the online method of classes in the future. As this work is among the very few studies conducted in Saudi Arabia which examines students' about online versus offline learning, it provides some information about teaching and learning preferences and perceptions. There is further need for larger, multicentric, nationwide studies offering a deeper understanding of the state of student

motivations, adaptations, coping mechanisms, and different learning needs during the ongoing pandemic.

**Author Contributions:** Conceptualization, F.R. and T.B.; methodology F.R., S.E.M. and M.T.A.; formal analysis, A.A.; investigation F.R. and T.B.; data curation, A.A.; writing—original draft preparation, S.E.M. and F.R.; writing—review and editing S.E.M. and M.T.A.; visualization, M.S.K. and A.A.A.-S.; supervision, A.A.S.; project administration, M.T.A. All authors have read and agreed to the published version of the manuscript.

**Funding:** The authors extend their appreciation to the Deanship of Scientific Research at King Khalid University, Saudi Arabia for funding this work through Small Groups Project under grant number RGP.1/62/43.

**Institutional Review Board Statement:** The study was conducted in accordance with the Declaration of Helsinki and approved by the Institutional Research Ethical Committee of the College of Medicine, King Khalid University for studies involving humans.

**Informed Consent Statement:** Informed consent was obtained from all subjects involved in the study.

**Data Availability Statement:** Not applicable.

**Acknowledgments:** We acknowledge the teaching assistants for helping us in data collection.

**Conflicts of Interest:** The authors declare no conflict of interest.

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
