# Peer review of "Students’ Preferences and Perceptions Regarding Online versus Offline Teaching and Learning Post-COVID-19 Lockdown"

_sustainability, doi:10.3390/su15032362_

Round 1

Reviewer 1 Report (Previous Reviewer 2)

the new version of the paper has provided answers for all the comments I made in the first iteration. 

The paper is ready for publication.

Anyway, I have a minor comment to be addressed: the references section makes use of several diverse fonts. Why? Please, fix it before publication.

Author Response

Thank you for your approval. As suggested the font issue has been fixed. 

Reviewer 2 Report (New Reviewer)

The paper aims to assess students' perceptions  and preference of online learning vs face-to-face learning after COVID-19 period. 

The study lacks theoretical foundation. The research objectives and the results should be grounded in learning science and psychological theories. The discussion should discuss and explain the results using well-known conceptual frameworks in e-learning, online learning, students’ motivation and self-regulation, etc. The results also should be discussed in relation to other similar studies and try to justify the differences, for example many items in the results show that students learn better in online mode, however there are many studies point out there are many challenges of students learning in online learning where students for example have less motivation. Also, the study finds that students in offline mode have less concertation comparing to face-to-face mode, this is also should be explained because instructor real presentation in classroom should help students concentrate more than online learning where students at home with many distractions like other family members, kids, etc.

There is no enough details about the survey like the whole structure, type of the questions...etc. It says there were 17 items; however, the results show more than that number. 

There are some questions not related directory to the study objective, however they might explain the results, for example fear of getting infected, challenges of transportation.

 The tables can be organized better to understand the contents easier.

The paper refer to student as "patient" in page 3. 

Author Response

The results also should be discussed in relation to other similar studies and try to justify the differences, for example many items in the results show that students learn better in online mode, however there are many studies point out there are many challenges of students learning in online learning where students for example have less motivation.

Reply: As suggested by the respected the results have been discussed in relation to other similar studies  and references 18, 34 and 46 have been added.

Also, the study finds that students in offline mode have less concertation comparing to face-to-face mode, this is also should be explained because instructor real presentation in classroom should help students concentrate more than online learning where students at home with many distractions like other family members, kids, etc. 

Reply: As suggested this has been mentioned. Please see page 11 lines 290-294

There is no enough details about the survey like the whole structure, type of the questions...etc. It says there were 17 items; however, the results show more than that number. 

Reply: This has been mentioned as pointed. Page 3 lines 128-129

The tables can be organized better to understand the contents easier.

Reply: Tables have been organised

The paper refer to student as "patient" in page 3. 

Reply: This has been corrected.

Reviewer 3 Report (New Reviewer)

This work is very appropriate in terms of its content and subject of interest.
Authors should complete standard ideas or terms throughout the paper, e.g., is COVID-19 than just place COVID infection or corona. Or "lockdown of institutions" due to COVID-19 pandemic (instead of COVID-19-lockdown or COVID-19 lockdown).

In section 2.3 of the methodology, the format of the second paragraph should be improved. 
Figures 1, 2 and 3 should be improved, review other finished works of the same journal or the indications of the editors, for example, the figures do not have title in the image, this title should be figure caption. The figure caption goes at the bottom. The size of the figures should be proportionally placed on the sheet. Compare figure 3 with number 1 and 2, they lack the same format, they should improve their presentation. Figure 3 should improve the use of color and differentiate the two categories (online and offline).
Table 1, should be proportionally located on the sheet. This table highlights the measurements.
Figures should go between paragraphs, i.e. the content of the figure should be explained between paragraphs and then explain the next figure. Improve the writing and placement of the figures and table.
References 15, 26-29, 31-33, 39, 43, 44 and 52 should be improved and revised due to the format used.
It is a good research work and this should be better presented.

Author Response

As suggested the following  or terms throughout the paper have been corrected,  e.g., is COVID-19 than just place COVID infection or corona. Or "lockdown of institutions" due to COVID-19 pandemic (instead of COVID-19-lockdown or COVID-19 lockdown).
In section 2.3 of the methodology, the format of the second paragraph has been improved. Figures 1, 2 and 3 have been improved. The figure caption is mentioned at the bottom. The size of the figures has been proportionally placed on the sheet. All figures have been improved in presentation. Figure 3 has been improved the use of color and differentiate the two categories (online and offline).
Table 1, has been proportionally located on the sheet. 
Figures have gone between paragraphs as suggested. 
All the References have been improved and revised as per  the format used.

Round 2

Reviewer 2 Report (New Reviewer)

Although the paper has some improvements, it  still  lacks linking these findings with the theory of learning science in terms of motivation, online learning...etc.

There are some typos  need to be fixed. For example:

Line 261: the reference [18[c], also the reference number should be before the dot .

Line 315 same as above.

Author Response

As suggested, the findings have been linked  with the theory of learning science in terms of motivation, online learning...etc. Please see references 19, 20, 42, 43, 44 and 62 have been added

Typo errors have been corrected as suggested

This manuscript is a resubmission of an earlier submission. The following is a list of the peer review reports and author responses from that submission.

Round 1

Reviewer 1 Report

The paper has a good structure. The research is interesting, important and relevant to this scientific area, nevertheless presents a minim contribution to the scientific knowledge.

The state-of-the-art presented is focused in COVID-19 and respective impacts in society, however, is crucial to understand the conceptual framework associated to the learning methods. What is understand by learning methods? What were applied before pandemic lockdown? ... Among others

The state-of-the-art should be improved accordingly.

The methodology needs to explain further details, namely:

-           the number of participants it is not clear (line 96 to 98) “A total of 175 forms 96 distributed, 95 to English students and 80 forms to finance students were distributed. A 97 total of 480 students returned completed forms-70 students from English and 50 students 98 from the finance department.” It is not comprehensible the number 480? If the math’s is correct is not well explained.

-           what were the data collect instruments? 

-           what was the research question? 

-           what were the hypotheses? 

-           what is the relationship between the data collect and the identification of the learning methods;

-           What scales were used to measure the Perceptions of the: tiredness, unhappy, among others;

After clarification of overall previous aspects, moreover this study is interesting to apply to students from different backgrounds/scientific areas crossing different framings.

The results and discussion should be presented and articulated with the previous suggestions introduced in the methodology. The conclusions should be aligned with previous improvements, the future perspective and the study limitations should adjusted.

Author Response

Reply to reviewers comments and suggestions

The paper has a good structure. The research is interesting, important and relevant to this scientific area, nevertheless presents a minim contribution to the scientific knowledge.

The state-of-the-art presented is focused in COVID-19 and respective impacts in society, however, is crucial to understand the conceptual framework associated to the learning methods.

Q.1 What is understand by learning methods?

Response:

Here we are concerned with the mode of Learning and teaching by which we deliver lectures for example traditional learning method (offline), online learning and blended method which means both offline and online mode of learning and teaching.

Reference:  Setyawan H. Blended Method: Online-Offline Teaching And Learning, On Students’ Reading Achievement. English Education: Jurnal Tadris Bahasa Inggris. 2019 Jul 5;12(1):22-33.

Online learning is described by most authors as access to learning experiences via the use of some technology.

Reference:  Carliner, S. (2004). An overview of online learning (2nd ed.). Armherst, MA: Human Resource Development Press.

Online learning has been identified as a more recent version of distance learning which improves access to educational opportunities for learners described as both nontraditional and disenfranchised.

Reference:  Benson, A. (2002). Using online learning to meet workforce demand: A case study of stakeholder influence. Quarterly Review of Distance Education, 3(4), 443−452.

Reference:  Conrad, D. (2002). Deep in the hearts of learners: Insights into the nature of online community. Journal of Distance Education, 17(1), 1−19.

As suggested by the respected reviewer the above changes have been incorporated in the revise manuscript. Please see Line 112-121.

Q.2 What were applied before pandemic lockdown?

Response:

Offline learning face to face method was applied before lockdown in the addressed institution (Added in Line 122-123)

The state-of-the-art should be improved accordingly.

Response: As suggested it has been improved.

The methodology needs to explain further details, namely:

  1. the number of participants it is not clear (line 96 to 98) “A total of 175 forms 96 distributed, 95 to English students and 80 forms to finance students were distributed. A 97 total of 480 students returned completed forms-70 students from English and 50 students 98 from the finance department.” It is not comprehensible the number 480? If the math’s is correct is not well explained.

Response: As suggested this has been corrected in methodology (Please see line 102-104)

-           Q. 3 what were the data collect instruments?

Response: Self-constructed questionnaire was utilized to collect the data based on personal student’s feedback. (Please see line 111-114)

-          Q. 4  what was the research question?

Response:  

The research question was: What are the student’s preferences and perceptions regarding the teaching-learning methods after the resumption of face to face offline teaching?  (line 107-109)

-           Q. 5 what were the hypotheses?

Response:  

We hypothesized that students continue to prefer the online teaching-learning methods post COVID-19 lockdown. (line 106-107)

-           Q. 6 what is the relationship between the data collect and the identification of the learning methods;

Response:  

Traditional mode of teaching and learning was implemented before and after lock down in the addressed institution, while during lockdown suddenly there was shift of traditional learning to online learning. (Line 125-127)

-           Q. 7 What scales were used to measure the Perceptions of the: tiredness, unhappy, among others;

Response:  

Only student’s subjective feelings about tiredness and being unhappy was considered in the study (line 129-130)

After clarification of overall previous aspects, moreover this study is interesting to apply to students from different backgrounds/scientific areas crossing different framings.

The results and discussion should be presented and articulated with the previous suggestions introduced in the methodology. The conclusions should be aligned with previous improvements, the future perspective and the study limitations should adjusted.

Response:  As suggested by the respected reviewer the discussion has been strengthened        References 14, 15, 18, 19, 30, 31, 32, 36, 37 and 38 have been added. Please see line 195-202, 217 to 219, 224 to 227, 228 to 234, 249 to 253, 265-268, 275-279 .

Reviewer 2 Report

Abstract: The structured abstract should be made emphasizing the structure. Having [...] Introduction: [...] and [...] Results: [...] with the same font and style as the text does not facilitate the reading (which is the goal of a structured abstract).

Introduction: major contributions of the article should be emphasized. Structure of the paper is absent. 

Lines 39-83: Please, fix the listing with an enumerate or itemize block. Or are those (<number X>) citations? In this case, I suggest the authors to check the format of citations in this journal because I think it should be actually different.

No related works section? Authors must add a related works section where they discuss and compare their results with those obtained by other similar studies carried in different universities all over the world (e.g.,https://doi.org/10.1007/978-3-031-15845-2_9, https://doi.org/10.12669/pjms.36.COVID19-S4.2766, https://doi.org/10.1002/jdd.12339, https://doi.org/10.3390/educsci10120355, etc.). By reading their work, all the interested ones should grasp a series of insights stemming both from this paper's results but also from the comparison with the others. This gives the overall pictures of covid-19 impact on students.

Figures: authors must provide high-quality figures. Figures also deserves a caption. Authors must add it for each figure. Furthermore, the authors must provide figures with the same sizing in the text to improve the sharpness of the paper.

Tables: please improve the readability of the tables using different font and styles.

Conclusion: this section is rather short and weak. Please improve it discussing more on the impact of the findings and so on.

Author Response

Abstract: The structured abstract should be made emphasizing the structure. Having [...] Introduction: [...] and [...] Results: [...] with the same font and style as the text does not facilitate the reading (which is the goal of a structured abstract).

Response:  

Introduction inserted at line 11

Methodology inserted at line 19

Results inserted at line 22

Conclusion inserted at line 32

Introduction: major contributions of the article should be emphasized. Structure of the paper is absent.

Response:  This has been revised as suggested.

This paper tries to elucidate the opportunities and challenges in higher education as regards the mode of teaching-learning: offline versus online. The findings of this study may be used to devise the future strategy for teaching-learning by building upon experiences of eLearning during the peak of the pandemic in order to develop a hybrid method of pedagogy which has the advantages of both the offline and online approaches of teaching-learning. (Line 92-95)

Lines 39-83: Please, fix the listing with an enumerate or itemize block. Or are those (<number X>) citations? In this case, I suggest the authors to check the format of citations in this journal because I think it should be actually different.

Response:  This has been revised as suggested.

No related works section? Authors must add a related works section where they discuss and compare their results with those obtained by other similar studies carri…

Response:  As suggested by the respected reviewer the discussion has been strengthened        References 14, 15, 18, 19, 30, 31, 32, 36, 37 and 38 have been added. Please see line 195-202, 217 to 219, 224 to 227, 228 to 234, 249 to 253, 265-268, 275-279 .

Round 2

Reviewer 2 Report

Authors have not provided a point to point response to my comments

Please re-consider the following ones who lack of a proper reply:

No related works section? Authors must add a related works section where they discuss and compare their results with those obtained by other similar studies carried in different universities all over the world: among the newest ones I suggest https://doi.org/10.1007/978-3-031-15845-2_9, https://doi.org/10.1152/ADVAN.00141.2021, https://doi.org/10.29333/pr/11551. By reading their work, all the interested ones should grasp a series of insights stemming both from this paper's results but also from the comparison with the others. This gives the overall pictures of covid-19 impact on students.

Figures: authors must provide high-quality figures. Figures also deserves a caption. Authors must add it for each figure. Furthermore, the authors must provide figures with the same sizing in the text to improve the sharpness of the paper.

Tables: please improve the readability of the tables using different font and styles.

Conclusion: this section is rather short and weak. Please improve it discussing more on the impact of the findings and so on.

Author Response

No related works section? Authors must add a related works section where they discuss and compare their results with those obtained by other similar studies carried in different universities all over the world: among the newest ones I suggest https://doi.org/10.1007/978-3-031-15845-2_9, https://doi.org/10.1152/ADVAN.00141.2021, https://doi.org/10.29333/pr/11551. By reading their work, all the interested ones should grasp a series of insights stemming both from this paper's results but also from the comparison with the others. This gives the overall pictures of covid-19 impact on students.

Reply: As suggested by the respected reviewer we have strengthened the discussion section.  Please see Reference number 14, 15, 17,18, 27, 28, 29 and 39  mentioned with green color in manuscript.

Figures: authors must provide high-quality figures. Figures also deserves a caption. Authors must add it for each figure. Furthermore, the authors must provide figures with the same sizing in the text to improve the sharpness of the paper.

Reply: We have added high-quality figures as suggested.

Tables: please improve the readability of the tables using different font and styles.

Reply: Revised as suggested

Conclusion: this section is rather short and weak. Please improve it discussing more on the impact of the findings and so on.

Reply: Conclusion has been strengthened as suggested.

Round 3

Reviewer 2 Report

The authors have improved a lot the paper. I appreciate their effort.

The research question (lines 114-115) can be put in a more visual appealing form. Maybe the authors can work on this.

Author Response

Dear Reviewer, Thank you for helping us improve our manuscript.  As per your kind suggestion, we have added figure 3 in the manuscript so that the research question appears in a more visual appealing form.